# Nodal is a short-range morphogen with activity that spreads through a relay mechanism in human gastruloids

Lizhong Liu [1], Anastasiia Nemashkalo[2], Luisa Rezende[1], Ji Yoon Jung[1], Sapna Chhabra[1,3], M. Cecilia Guerra[1], Idse Heemskerk [4,5] & Aryeh Warmflash [1,6✉]

Morphogens are signaling molecules that convey positional information and dictate cell fates during development. Although ectopic expression in model organisms suggests that morphogen gradients form through diffusion, little is known about how morphogen gradients are created and interpreted during mammalian embryogenesis due to the combined difficulties of measuring endogenous morphogen levels and observing development in utero. Here we take advantage of a human gastruloid model to visualize endogenous Nodal protein in living cells, during specification of germ layers. We show that Nodal is extremely short range so that Nodal protein is limited to the immediate neighborhood of source cells. Nodal activity spreads through a relay mechanism in which Nodal production induces neighboring cells to transcribe Nodal. We further show that the Nodal inhibitor Lefty, while biochemically capable of long-range diffusion, also acts locally to control the timing of Nodal spread and therefore of mesoderm differentiation during patterning. Our study establishes a paradigm for tissue patterning by an activator-inhibitor pair.

[1] Department of Biosciences, Rice University, Houston, TX, USA. [2] Los Alamos National Laboratory, CINT/B11 Division, Los Alamos, NM, USA. [3] Developmental Biology Unit, EMBL Heidelberg, Heidelberg, Germany. [4] Department of Cell and Developmental Biology, University of Michigan Medical School, Ann Arbor, MI, USA. [5] Department of Physics, University of Michigan, Ann Arbor, MI, USA. [6] Department of Bioengineering, Rice University, Houston, TX, USA. ✉email: aryeh.warmflash@rice.edu

One of the central questions in developmental biology is how spatial distributions of extracellular morphogens are controlled and interpreted to produce patterns of cell fates. While it is commonly believed that positional information is conveyed by long-range gradients of morphogens[1,2], direct evidence is difficult to obtain as they are present in the extracellular space and often effective at low (nanomolar) concentrations[3]. Some recent studies have suggested that morphogens can be short range, and that extracellular diffusion of some morphogens may be dispensable for normal development[4–6].

Nodal, a secreted member of the transforming growth factor ß (TGF-ß) family, is essential for stabilization of the epiblast state, induction of both the anterior-posterior and left-right axes, and specification of mesoderm and endoderm[7,8]. Nodal signals by binding to a receptor complex comprised of type I and type II TGF-ß receptors and a coreceptor, known as Crypto[9]. Genetic studies have shown that Nodal signals as a heterodimer with TGF-ß family members GDF1/3 (Vg1 in Zebrafish)[10–13]. The activated receptor complex phosphorylates signal transducers Smad2/3 at their C-terminal which in turn bind to cofactor Smad4 and form a larger complex which regulates target genes in the nucleus[14].

A differential diffusivity model has been proposed to account for the creation of a Nodal gradient through interactions with Lefty1 and Lefty2 (hereafter Lefty1/2). Nodal signaling activates both its own expression and that of Lefty1/2. Lefty1/2 diffuse farther than Nodal so that local autoactivation creates territories of high signaling activity, while long range inhibition suppresses signaling at a distance[15]. Evidence from ectopic expression experiments performed in Zebrafish supports this model[15–17], however, overexpression studies can saturate receptors and other binding sites, altering the spatial range of secreted molecules. In addition to paracrine signaling over several cell diameters, other models which do not rely on extracellular transport, including a relay model in which signaling induces the transcription of the gene encoding the morphogen in neighboring cells have also been suggested to explain Nodal signaling in the early vertebrate embryo[5,18–21]. Direct validation has been prevented by the difficulty of measuring endogenous morphogens during patterning. Moreover, in Xenopus and Zebrafish where these issues were studied, there are multiple Nodal genes which signal at different ranges[15–17,19,22], while there is only one mammalian Nodal gene. This provides an opportunity to study the ligand distribution for this pathway by studying the localization of a single protein. To date, owing to technical limitations, endogenous Nodal protein has never been visualized, therefore its spatiotemporal distribution in mammalian embryos remains a mystery.

Here we visualize and quantitatively measure the levels of endogenous Nodal and Lefty1/2 mRNA and protein in human embryonic stem cells (hESCs) and in a model of human germ layer patterning, micropatterned hESC-based (2D) gastruloids[23,24]. In this model, exogenous BMP signaling triggers a cascade of signaling through the BMP, Wnt, and Nodal pathways causing germ layer differentiation. These experiments revealed that Nodal protein and signaling activity are restricted to cells that immediately touch producing cells. Through genetic manipulation, we show that propagation of the Nodal signal requires that the Nodal gene be intact in receiving cells, suggesting a transcriptional relay model. As this relay causes Nodal activity to spread inwards in the gastruloid, Lefty is transiently expressed specifically at the signaling front. Deletion of both Lefty genes leads to earlier and more rapid spreading and an expansion of mesendoderm differentiation. This suggests that the function of Lefty during gastrulation is to control the timing and spread of the Nodal signaling wave to properly induce mesendoderm.

## Results

### Visualization of fully functional endogenous Nodal protein.
To visualize endogenous Nodal protein, we generated hESCs with an mCitrine::Nodal (hereafter, cNodal) fusion allele. In these cells, the fluorescent mCitrine is inserted between the Nodal pro-domain and mature domain (Supplementary Fig. 1a, b). After being translated, the full-length protein will be cleaved by convertases[25], splitting the pro-domain from the mCitrine-tagged mature domain. We generated both heterozygous Nodal$^{+/Cit}$ and homozygous Nodal$^{Cit/Cit}$ cell lines. Unless specifically noted otherwise, all results reported are for homozygous Nodal$^{Cit/Cit}$ cells. The cNodal protein was readily visualized in modified cells where Nodal could be found in the cytoplasm and on the basolateral side of the cells with less protein localized to the apical side (Fig. 1a). We also observed occasional long cell protrusions containing cNodal protein (Supplementary Fig. 2).

We found that mCitrine tagging does not compromise hESCs pluripotency (Supplementary Fig. 1c), and natural regulation of Nodal expression is preserved (Fig. 1b). In particular, cNodal expression level increases in response to addition of Activin A, a ligand of Activin/Nodal signaling, and Wnt3A, an upstream signal (Fig. 1b, c, Supplementary Movie 1–2). In line with previous studies, BMP4 does not stimulate Nodal expression directly (Fig. 1b), however, experiments in gastruloids confirm that BMP signaling induces Nodal through a Wnt intermediate, as with untagged Nodal (Fig. 1c, Supplementary Movie 3)[26,27].

We utilized cNodal hESCs to make gastruloids. We found that cNodal heterozygotes and homozygotes give rise to normal fate patterns indistinguishable from wild type (WT) patterns (Fig. 1d, Supplementary Fig. 4a, b), while gastruloids made with Nodal knockout hESCs lack the mesodermal layer[28] (Supplementary Fig. 3a), consistent with the phenotype of Nodal$^{−/−}$ mice[7]. Moreover, Nodal$^{Cit/Cit}$ gastruloids show normal patterns of activated Smad2/3 (Fig. 1e, Supplementary Fig. 3b), while we have previously observed significantly lower levels in Nodal$^{−/−}$ cells[28]. Together, these results demonstrate that the cNodal fusion is fully functional.

### Nodal signaling activity propagates by a relay mechanism.
Next, we surveyed the ranges of Nodal and Lefty proteins through juxtaposition of cNodal cells (senders) with Lefty1/2 null cells (receivers) (Fig. 2a–c, Supplementary Figs. 5 and 6)[29]. Activin induced senders produce both Lefty and cNodal, while receiver cells do not produce either. Therefore, the spatial range of the two proteins can be obtained via measuring fluorescence intensity in the receivers as a function of the distance from the senders. We failed to observe Nodal protein beyond the receiver cells immediately adjacent to the senders (Fig. 2b, c, e and Supplementary Fig. 6b), while Lefty traveled over 6–8 cell tiers from the source and maintained stable gradients over time (Fig. 2c and Supplementary Fig. 6b). Although cNodal does not spread over a distance of more than one cell diameter, extracellular cNodal can be captured by a GFP (mCitrine)-specific nanobody (hereafter CitrineTrap) presented on surface of neighboring receiving cells. The CitrineTrap consists of an extracellularly presented GFP nanobody, which also recognizes mCitrine, the mouse CD8 transmembrane domain and an intracellular mCherry[30]. We found that when cNodal cells were induced with Activin to express high levels of Nodal and either mixed with or juxtaposed to CitrineTrap cells, cNodal was enriched on the surface of the CitrineTrap cells immediately adjacent to cNodal cells, while a mCitrine fused to intracellular domain of E-cadherin could not be captured by CitrineTrap (Fig. 2d, e, Supplementary Fig. 7). After withdrawing of Activin, cNodal levels decrease in the source cells, however, CitrineTrap cells still display high levels of cNodal

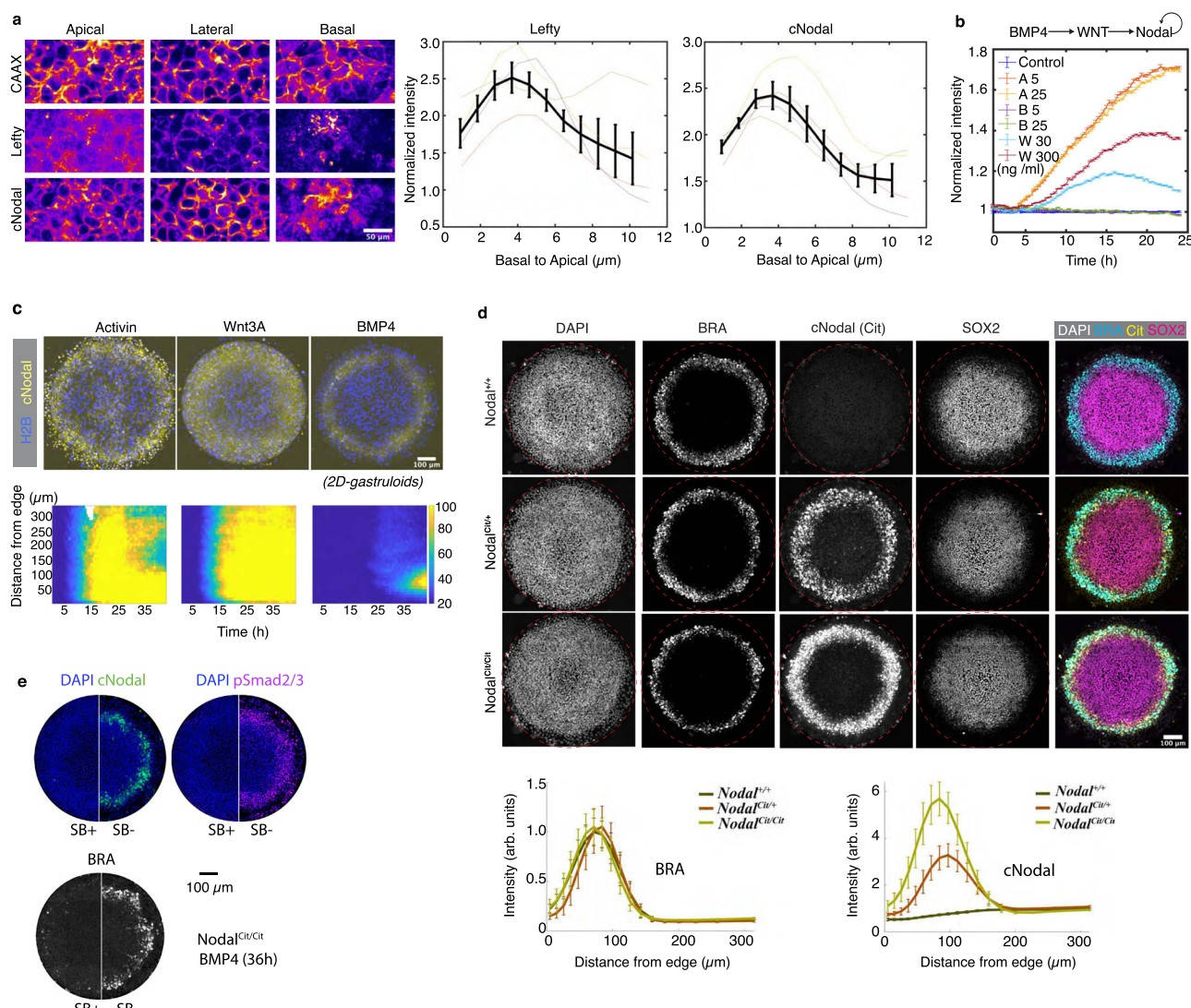

**Fig. 1 Visualization of a fully functional endogenous Nodal protein in hESCs and 2D-gastruloids. a** Immunostaining of cNodal (anti-GFP antibody) and Lefty (anti-Lefty) in Activin treated Nodal$^{cit/cit}$ cells with mCherry-CAAX membrane marker. Both cNodal and Lefty showed the highest levels at the basolateral side (normalized to mCherry-CAAX). Data are presented as mean values +/− SEM, $n = 4$ biological replicates. Scale bar 50 μm. **b** Quantification of cNodal expression in response to addition of Activin (5, 25 ng/ml), Wnt3A (30, 300 ng/ml), or BMP4 (5, 25 ng/ml). Normalized mean intensity of cNodal from time-lapse imaging over 24 h with a 20-minute interval is quantified. Data are presented as mean values +/− SEM, $n = 5$ biological replicates. **c** Activin, Wnt3A, or BMP4 treated micropatterned cNodal hESCs with CFP-H2B nuclear marker. (bottom) Mean intensity of cNodal from time-lapse imaging over 43 h with 30-minute interval is quantified as function of time and distance from colony edge. **d** (left) Representative images of 2D-gastruloids with the indicated genotypes are shown. (right) Quantification of BRA and cNodal levels in 2D-gastruloids with the indicated genotypes after 40 h of BMP4 treatment. Data are presented as mean values +/− SEM, $n = 10$ independent colonies. Mean intensity and standard error across colonies are plotted as a function of distance from colony edge. Colonies for imaging were chosen based on visualization of DAPI without visualizing the other channels. Brachyury (BRA) represents mesodermal fate. SOX2 represents ectodermal fate. Cit represents anti-mCitrine (GFP antibody). **e** SB-431542 an ALK4/5/7 inhibitor was used to inhibit Nodal signaling in homozygous cells and the expression of cNodal, pSmad2/3 and BRA were compared (representative of $n = 6$ colonies for each condition). Scale bar, 100 μm. Maximum intensity projection images are shown in all figures.

captured on the membrane. These results directly confirm that cNodal is secreted and passed to neighboring cells.

The failure of Nodal protein to spread from source cells seemingly contradicts our previous observations that Nodal signaling activity can propagate from the edge to the center of gastruloids, even in the absence of upstream Wnt signaling[28,31]. This discrepancy is potentially consistent with the idea that signaling spreads primarily through a relay mechanism in which Nodal activity induces transcription of Nodal ligand in immediately adjacent cells (Fig. 3a).

To test the relay hypothesis (Fig. 3a), juxtaposition experiments with various combinations of sender and receiver hESCs were carried out (Fig. 3b). In particular, we induced Nodal expression in

sender cells by treating them with Activin and then placed them adjacent to either wild type (WT) or Nodal$^{−/−}$ (NKO) cells. As controls, we also used non-induced cells or induced Nodal$^{−/−}$ cells (NKO) as senders. Dramatically elevated levels of Smad2/3 nuclear accumulation were only observed when both the sender and receiver cells had a functional Nodal protein, either Nodal$^{+/+}$ or Nodal$^{Cit/Cit}$ (Fig. 3b). When Nodal$^{−/−}$ cells were used as receivers, the signal was not transmitted beyond the receiving cells which touch the senders, suggesting that a relay involving induction of Nodal ligand in the receivers is required. Non-induced and Nodal$^{−/−}$ cells also failed to induce a response, showing that the response is specific to Nodal from the senders.

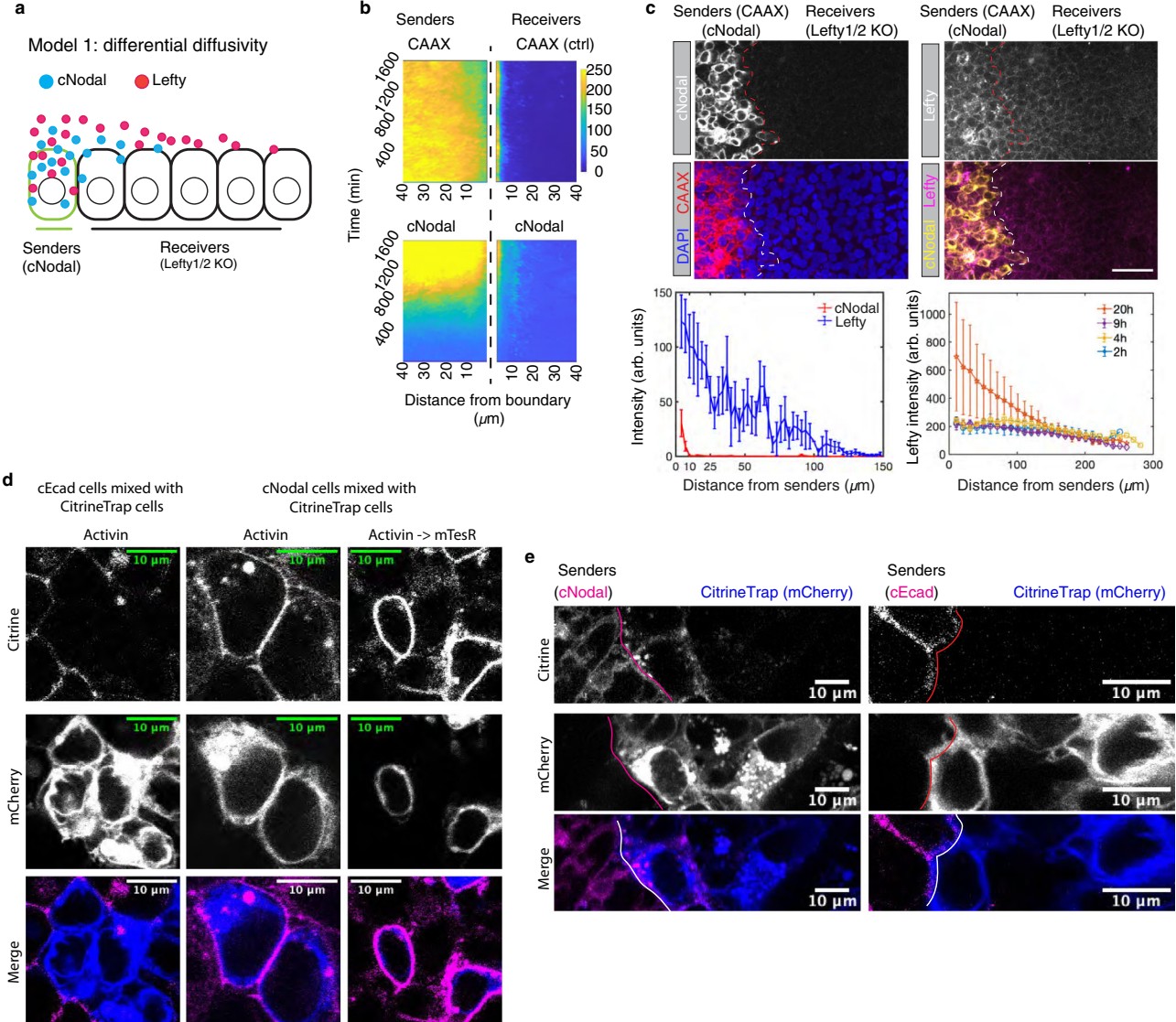

**Fig. 2 Nodal only reaches cells that touch producing cells. a** Schematic of protein differential diffusivity model. Senders (green outline) represent cells that produce cNodal (solid blue circles) and Lefty1/2 (solid magenta circles) upon Activin induction. Receivers represent Lefty1/2 KO cells that cannot make either Lefty or cNodal proteins. **b, c** Experimental test of the differential diffusivity model. Sender cells are labeled by cell membrane localized mCherry-CAAX. **b** Live-cell imaging over 27 h with a 20-minute interval. mCherry-CAAX and cNodal levels in senders or receivers are quantified as function of distance from border. mCherry-CAAX in senders is used as a non-diffusive control. **c** Anti-GFP that recognizes mCitrine and anti-Lefty antibodies (Supplementary Fig. 5) were used to measure protein ranges. Quantitative analysis of staining images is shown. Mean intensity of cNodal (anti-GFP) and Lefty at 20 h (bottom left) or of Lefty at 2, 4, 9, 20 h (bottom right) is quantified as a function of distance from border ($n = 4$). Scale bar, 100 µm. Images are 20 h post juxtaposition. Data are presented as mean values $+/-$ SEM, $n = 4$ biological replicates. **d** Representative cells from a larger field of coculture of (left) E-Cadherin-mCitrine (cEcad) or (middle and right) cNodal cells with CitrineTrap cells. The middle panel only shows two CitrineTrap cells, right panel shows two CitrineTrap cells (blue) and surrounding cNodal cells. Scale bar, 10 µm. Similar results were observed in more than 3 repeated experiments. **e** Representative cells of juxtaposition experiments with induced cNodal cells or cEcad cells as senders and CitrineTrap cells as receivers. Experiments were repeated independently 3 times with similar results. Scale bar, 10 µm.

We also measured the induction of Nodal and Lefty transcripts in a similar assay (Fig. 3c, d, Supplementary Fig. 6c–e). Induced WT (iWT) senders were juxtaposed with non-induced WT cells or with Nodal$^{-/-}$ cells for up to 20 h. In the former case (iWT - WT), we observed gradients of Nodal RNA (Fig. 3c, d, Supplementary Fig. 6d), Lefty RNA and nuclear Smad2/3 (Supplementary Fig. 6e), which peaked at the border and declined with distance from the senders. However, when Nodal$^{-/-}$ cells were used as receivers, we did not observe elevated signaling activity or increased Nodal or Lefty mRNA production in the receivers (Fig. 3c, d, Supplementary Fig. 6d–e). Interestingly, lower levels of Nodal RNA and Lefty RNA were observed in the sender cells when juxtaposed with Nodal$^{-/-}$ cells, and by 20 h, induction of these Nodal targets was largely absent in the senders juxtaposed to Nodal$^{-/-}$ cells but maintained in senders juxtaposed to WT cells. These results indicate that propagation of signal is bidirectional and that signaling from the receivers is required to maintain Nodal expression in the senders. Together, these results support that a transcriptional relay is required for the propagation of Nodal signaling in space.

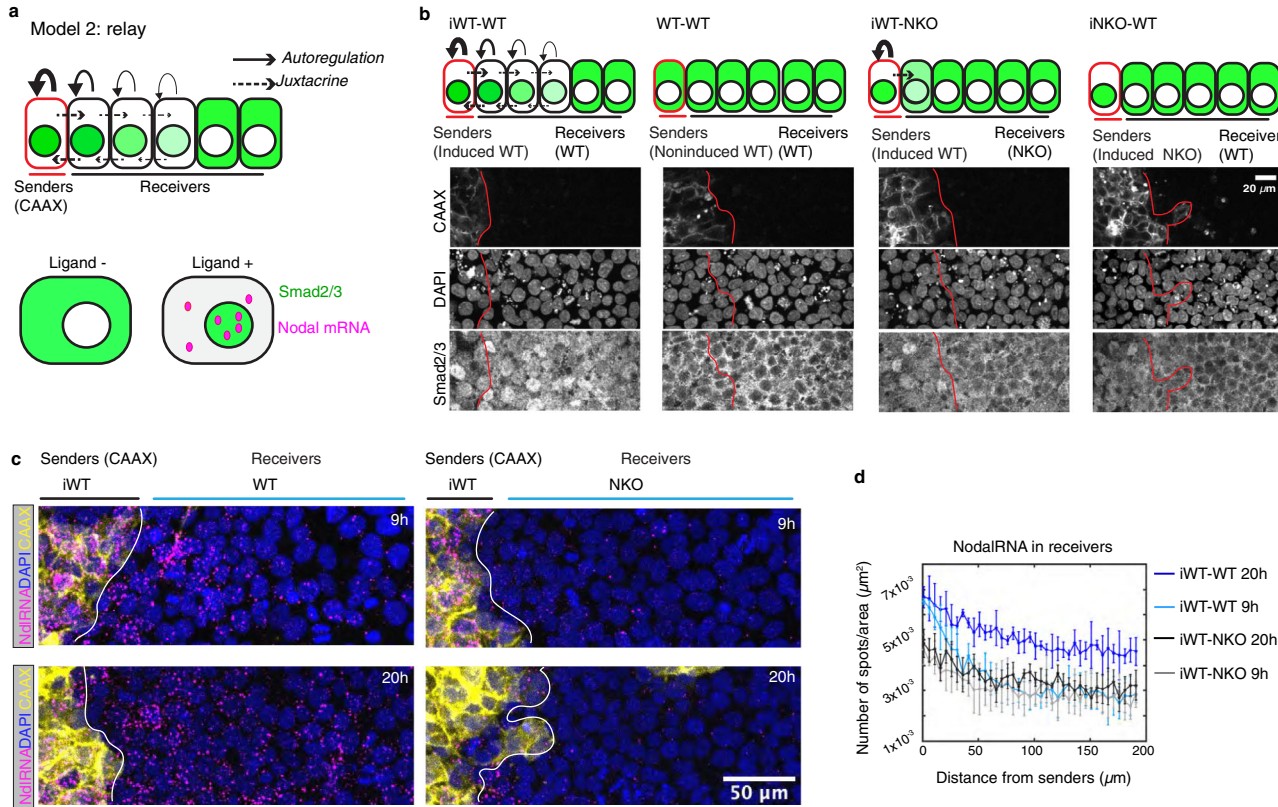

**Fig. 3 Nodal propagates by a relay mechanism. a** Schematic of relay model. Senders (red outline) represent induced cells that express Nodal. Solid line with arrowhead, represents Nodal positive autoregulation. Dashed line with arrowhead represents juxtacrine induction by Nodal. Nodal/Activin signaling transducers Smad2/3 (green) are located in cytoplasm in the absence of ligands. In the presence of Nodal or Activin, Smad2/3 translocate to nucleus, and Nodal transcription is upregulated. Nodal mRNA is shown as magenta solid circles. **b–d** Experimental test of the relay model. **b** Staining of Smad2/3 to measure Nodal signaling activity in juxtaposition experiments with cells with the indicated genotypes. Signal is nuclear when the pathway is activated. Sender cells in all conditions were labeled by mCherry-CAAX. Images in each case taken after 20 h juxtaposition $n = 4$, scale bar, 20 μm. **c**, **d** Experiments performed as in (**b**) and Nodal mRNA (NdlRNA) was measured. Nodal RNA was quantified as a function of distance from border ($n = 4$ for each condition) (**c**) Representative images of 9 h and 20 h post-juxtaposition is shown, scale bar, 50 μm. **d** Nodal RNA was measured in a separate experiment with 9 h and 20 h post-juxtaposition. iWT means induced wild type cells, iNKO means induced Nodal KO cells. Data are presented as mean values +/− SEM, $n = 4$ biological replicates.

**Both Nodal and Lefty function at short range during gastruloid patterning**. We then sought to test whether this hypothesis stands true in micropatterned human gastruloids. In this model, self-organized patterns are formed with extraembryonic cells at the colony edge, ectoderm at the center, and rings of mesendoderm in between. We first assessed Nodal and Lefty transcription in space and time (Fig. 4a, b). The results showed that peak expression of both Nodal and Lefty mRNA was first detected at the colony edge, then both expanded towards the colony center as development of the gastruloids proceeded. Although both moved inwards, Nodal mRNA expanded to cover a broader range while remaining high at its point of initiation, consistent with measurements of Smad2 signaling activity (Figs. 4c, d, 5a, Supplementary Fig. 8a–c and [28],[31]). In contrast, Lefty RNA, a direct Smad2-dependent Nodal signaling target, shifted rapidly inward so that it was only expressed transiently at each spatial position, and the peak active position moved continually inward. These results show that Nodal activity spreads inwards to cover a domain of increasing size driven by expression of Nodal mRNA, while Lefty mRNA is only expressed at the front of this spreading domain in the cells which have most recently activated Nodal signaling. These data suggest that Lefty expression is adaptive, increasing upon initial exposure to Nodal and then returning to low levels despite continued Nodal protein expression and signaling activity (Supplementary Fig. 10).

Next, by simultaneously tracing Nodal mRNA, Nodal protein and phosphorylated Smad2/3 (pSmad2/3) (Fig. 4c, d), we found that, in line with the relay model, Nodal protein and activity were only found in the region of Nodal mRNA production, suggesting essentially no transcriptionally-independent spread (Fig. 4d, Supplementary Fig. 8c). Interestingly, in contrast to its broader range in the juxtaposition assay, Lefty protein moved as a traveling wave, closely mirroring the mRNA wave (Fig. 4e, f, Supplementary Figs. 4c–d, 8d), suggesting that in this context, Lefty remains close to its source and moves inwards primarily through a shift in the region of transcription of Lefty mRNA. The difference between these assays may reflect the different time scales involved. In the juxtaposition assay, the Lefty gradient forms over 20 h (Fig. 2c), while in the gastruloid model, transient expression at each spatial position might not allow for spread of Lefty. Taken together, the results indicate that shifting patterns of Nodal and Lefty reflect a transcriptional relay, while extracellular movement of these proteins plays little to no role in patterning micropatterned gastruloids.

**Lefty restricts Nodal in space and time to control mesodermal patterning**. To better understand the function of Lefty, which was specifically expressed at the front of signaling, we created Lefty1/2 compound knockout (Lefty1$^{-/-}$, Lefty2$^{-/-}$, hereafter Lefty1/2$^{-/-}$, L1/2KO) hESCs. We confirmed loss of both alleles of Lefty1 and

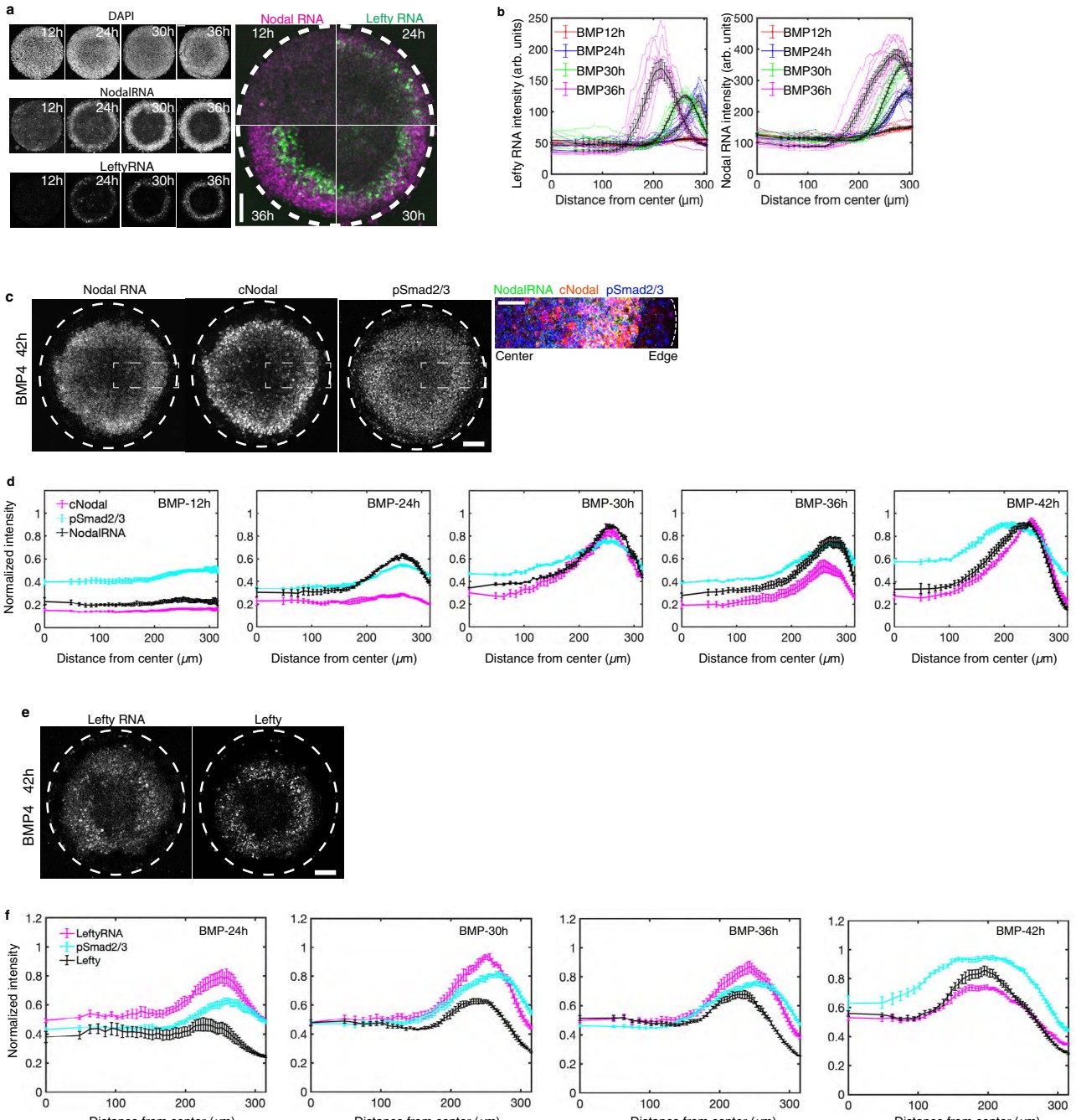

**Fig. 4 Nodal and Lefty spread via a relay mechanism in 2D-gastruloids. a** smFISH results showing mRNA for Nodal and Lefty1/2 at 12 h, 24 h, 30 h, 36 h post-BMP4 treatment, scale bar, 100 μm. **b** Quantitative analysis of the smFISH mean intensity as a function of distance from colony center at the indicated times, individual colonies are shown as colored line. Black line shows mean values +/− SEM, $n = 10$ biological replicates. **c** Representative images of gastruloids at 42 h, showing distribution of RNA and protein of Nodal and pSmad2/3, scale bar, 100 μm. The rectangular box indicates the enlarged area in the right panel, scale bar, 50 μm. **d** Simultaneous quantification of Nodal RNA and protein and pSmad2/3. Normalized mean intensity is quantified over time as function of distance from colony center. Data are presented as mean values +/− SEM, $n = 6$ biological replicates, quantification of individual colonies is shown in Supplementary Fig. 8c. **e** Representative images of gastruloids at 42 h, showing Lefty mRNA and protein, scale bar, 100 μm. **f** Quantification of Lefty RNA and protein over time, mean intensity as function of distance from colony center. Data are presented as mean values +/− SEM, $n = 6$ biological replicates, quantification of individual colonies is shown in Supplementary Fig. 8d.

Lefty2 via sequencing, and verified via immunofluorescence that Lefty protein was undetectable in these knockout cells (Supplementary Fig. 5b, c). The latter result confirmed loss of both Lefty1 and Lefty2, as siRNA experiments confirmed that the antibody detects both Lefty proteins (Supplementary Fig. 5a). Deletion of both Lefty genes did not compromise pluripotency, as reflected by the

expression of pluripotency markers Sox2, Oct4 and Nanog (Supplementary Fig. 1c).

We created gastruloids with WT hESCs or Lefty1/2$^{-/-}$ hESCs (Fig. 5). The results show that Lefty protein ablation led to upregulated Nodal production at 12 h with high levels peaking at the colony edge and extending in a shallow gradient towards the

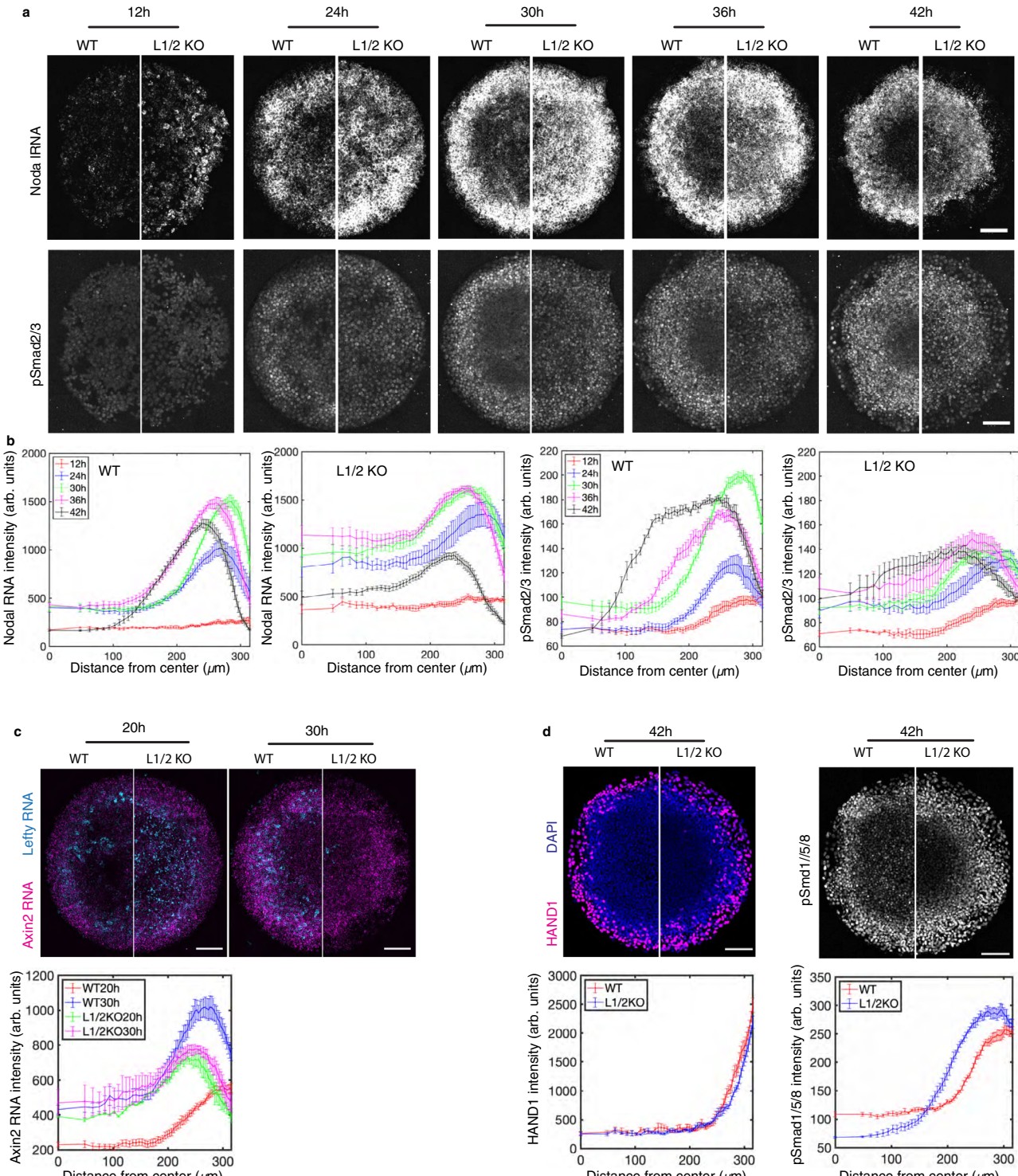

**Fig. 5 Lefty is needed to slow the spread of Nodal. a** Comparison of Nodal mRNA expression and pSmad2/3 over time in gastruloids created with WT and Lefty1/2 compound KO hESCs. Representative images at all time points are shown, scale bar, 100 μm. **b** Quantification of 6 colonies with indicated genotypes at each time point, data are presented as mean values $+/-$ SEM. **c** Analysis of Wnt signaling. Transcripts of Axin2, a Wnt signaling transcriptional target, and transcripts of Lefty1/2 in gastruloids at 20 h and 30 h were co-examined via smFISH. Mean intensity of FISH signal was quantified as function of distance from colony center. Data are presented as mean values $+/-$ SEM, $n = 6$ biological replicates. Scale bar, 100 μm. **d** Analysis of the BMP signal transducers pSmad1/5/8 and the BMP target HAND1 in 2D gastruloids at 42 h. Mean intensity is quantified as a function of distance from colony center. Data are presented as mean values $+/-$ SEM, $n = 6$ biological replicates. Scale bar, 100 μm.

center from 24 h to 42 h. In contrast in WT gastruloids, Nodal transcripts did not emerge at elevated levels until 24 h and then peaked at colony edge with fronts gradually propagating inwards (24–42 h). In line with Nodal expression patterns, compared to

WT gastruloids, pSmad2/3 levels were detected earlier and in a broader range extending to the colony center of Lefty1/2$^{-/-}$ gastruloids (Fig. 5a, b). In addition, analysis of Lefty transcripts shows that sustained pSmad2/3 levels did not stimulate sustained

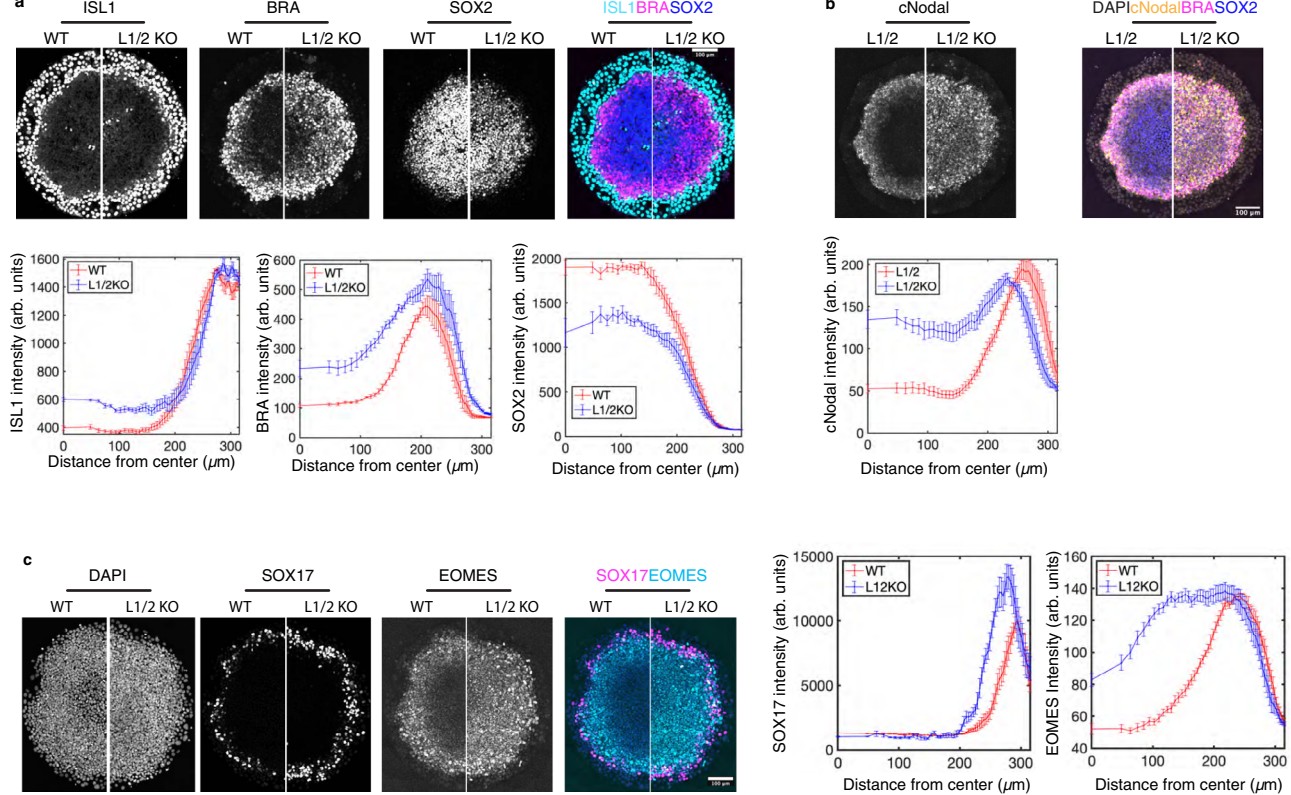

**Fig. 6 Lefty is needed to define the region of mesodermal differentiation. a** Comparison of ESI (WT) cells and ESI-Lefty compound knockout (L1/2KO) cells. Immunofluorescence for ISL1, extraembryonic marker, BRA, mesodermal marker, and SOX2, ectodermal marker. Quantification of 6 colonies for each condition. Data are presented as mean values +/− SEM. **b** Comparison of Nodal^Cit/Cit with intact Lefty1/2 (L1/2) and Nodal^Cit/Cit -Lefty compound knockout (L1/2KO) cells. Quantification of 6 colonies of each condition. Data are presented as mean values +/− SEM. **c** Comparison of ESI (WT) cells and ESI-Lefty compound knockout (L1/2KO) cells. Immunofluorescence for SOX17, definitive endodermal and PGC marker, and EOMES, mesendodermal marker. In all cases, normalized mean intensity is quantified as a function of distance from colony center, data are presented as mean values +/− SEM, $n = 6$ biological replicates for each condition. Images in (**a**, **c**) were acquired after 40 h of BMP4 treatment, images in (**b**) were acquired at 37 h. Scale bar, 100 µm in all panels.

Lefty transcription (Supplementary Fig. 10b), which supports the hypothesis discussed above of adaptive Lefty expression.

Previous studies have shown that a sequence of signaling activities through the BMP, Wnt, and Nodal pathways are responsible for patterning the gastruloids[28,32]. To verify whether or not the altered Nodal signaling is directly caused by Lefty depletion or might also involve changes in upstream signaling, we assessed BMP4 and Wnt signaling in Lefty knockout gastruloids. The results show Wnt signaling largely remain unchanged, while at 42 h there is an expansion of the BMP transducer pSmad1 but no corresponding change in the extraembryonic territory which expresses Hand1, a BMP target (Fig. 5c, d). Therefore, altered Nodal signaling is not likely caused by perturbation to upstream pathways. These results together suggest that Lefty is required to restrict Nodal signaling in space and time.

We then further investigated the role of Nodal-Lefty self-organization in fate patterning by surveying germ layer differentiation in control and Lefty1/2 KO gastruloids (Fig. 6). The data show that Lefty depletion results in expanded mesodermal (BRA+) fates at the expense of ectodermal cells (SOX2+) (Fig. 6a). In addition, we used CRISPR/Cas9 to deplete Lefty1/2 expression in cNodal cells and verified the loss of Lefty1/2 via sequencing and immunofluorescence (Supplementary Fig. 5d, e). Consistent with the smFISH data, we observed expanded cNodal expression coincident with expanded mesodermal (BRA+) fates (Fig. 6b). The mesendoderm marker EOMES was also dramatically expanded in Lefty1/2$^{-/-}$ cells (Fig. 6c). Interestingly, the range of SOX17 expression is unchanged

between WT and Lefty1/2 KO (Fig. 6c). A recent study showed that while SOX17 is often used as an endodermal marker, at this stage, most SOX17 positive cells represent primordial germ cells (PGCs), and that while these cells require Nodal signaling, the number of PGCs is not increased by higher levels of Nodal[33]. Our results here are consistent with these observations as upregulated Nodal via Lefty1/2 KO did not change the SOX17 expression domain. Mesendodermal differentiation was also detected earlier and more broadly in Lefty1/2 mutant gastruloids with BRA expression observed in the center where ectoderm usually forms at 24 h in Lefty1/2 mutant gastruloids, but not in WT gastruloids (Supplementary Fig. 9), suggesting that the early upregulation of Nodal signaling in this region drives rapid differentiation.

## Discussion

Here we have observed endogenous Nodal and Lefty in hESCs and during 2D gastruloid patterning. Visualization of protein together with the encoding mRNA allowed us to determine the range of action of these molecules. Nodal does not travel more than one cell diameter in any context we examined, and, Lefty protein expression, while capable of forming a stable gradient, closely mirrors that of Lefty mRNA during gastruloids patterning. Nodal moves by a transcriptional relay mechanism, and patterns of transcription largely account for both Nodal and Lefty spreading. As Nodal expands, Lefty is expressed specifically at the front of expansion, and compound deletion of Lefty1/2 causes

more rapid inward spread and an expansion of the mesendo-dermal territory, arguing that Lefty is crucial for controlling the timing of Nodal propagation and in turn the extent of differentiation.

Relatively early experiments using overexpression in Xenopus showed that TGF-ß family members can have different signaling ranges which are at least partially regulated by the prodomain[22,34–36]. For example, fusing the pro-domain of Activin to a Xenopus Nodal gene (Xnr2) increased its signaling range[22]. Interpretation of these experiments is complicated by the use of overexpressed protein, and conflicting results were sometimes obtained, for example, some groups found Xnr2 range to be limited to one or two cell diameters while others suggested it could signal over a long distance[22,35]. More recently, experiments in Zebrafish have shown that Nodal signaling is limited to only a few cell tiers from the embryonic margin, while over a longer range, some of the same mesodermal targets are activated by FGF[5,21]. Moreover, it was argued that since expression of Nodal overlaps with that of a Nodal reporter, Nodal must be short range[5]. However, the fact that Nodal is its own target could confound interpretation of these results, as this would naturally produce overlap between Nodal expression and that of other Nodal targets or reporters. Visualizing endogenous Nodal protein provides a more direct assay for addressing these questions, and existence of only a single Nodal gene in mammals, together with the strong phenotype of its deletion both in mice and in human gastruloids[28,37], facilitates analysis. Our observations of the signaling range of endogenously expressed cNodal together with the finding that Nodal$^{-/-}$ cells cannot propagate the Nodal signal at all, strongly argue that the signaling range of mammalian Nodal in this context is limited to one cell.

Studies in Zebrafish suggest that the range of Nodal signaling is limited through its capture by the coreceptor Cripto[38]. Although embryos deficient for Cripto cannot signal, Nodal protein from the margin can disperse over a much wider range than in wild-type embryos. Whether such a mechanism operates in mammalian embryos or gastruloids is unknown, but the fact that Cripto is capable of functioning non-cell autonomously in these systems argues against it playing this role[39]. If Cripto functions differently in mammals, this raises the question of what limits the spread of Nodal to a single cell diameter.

Our recent work argues against Nodal and Lefty forming a self-organizing Turing system, because the patterns of signaling activity do not have a fixed length scale as would be expected[28,31]. Instead, Nodal signaling eventually evolves to a pattern with high levels throughout the colony except at the edge where activity is lower. Mesendodermal differentiation does not cover this whole range indicating that other parameters besides the level of Nodal activity are important for setting the boundaries of this territory. The lack of Nodal spreading observed here is different from the typically conceived picture, however, it could still be consistent with a Turing instability if Nodal propagated slowly or at short range through a relay mechanism, while Lefty spread at longer range. However, we observe that Lefty proteins do not appear to function at long range during patterning, and therefore would not satisfy the requirements for long range inhibition. Thus, the present study adds to the evidence against a Turing mechanism for mammalian germ layer patterning.

There are several possible explanations for the differences in Lefty range between juxtaposition experiments in which they traverse multiple cell diameters and 2D gastruloids where they remain close to the source cells that produce them. As noted above, a key difference is that of time scale: long-term expression of Lefty in the sender cells in the juxtaposition experiments allows for a Lefty gradient to form, while the rapidly evolving patterns of Lefty transcription in 2D gastruloids might not. A second possibility is that the higher cell density or other factors such as additional extracellular matrix proteins might be present in the 2D gastruloids but not in juxtaposition experiments and could hinder the spread of Lefty. Finally, the available FISH probes and antibodies do not distinguish between Lefty1 and Lefty2. It is possible that these have different ranges of action and that there is a different balance of Lefty1/2 expression in the gastruloids compared to the sender cells in the juxtaposition experiments. Distinguishing between these non-mutually exclusive possibilities is an interesting topic for future study.

Remarkably, in Zebrafish, deletion of both Lefty genes can be rescued by incubation of the embryos with a small molecule signaling inhibitor suggesting that neither feedback inhibition nor restricted spatial localization is required for the effects of Lefty on patterning[40]. This would seem to suggest that Lefty is needed simply to lower overall Nodal signaling levels, however, signaling activity has not been directly measured in these cases, and it may be that both Lefty and the small molecule inhibition also slow down the speed of signal propagation. Our data here showing earlier spread of Nodal when Lefty genes are deleted are more consistent with this second possibility. It would be interesting to perform such signaling measurements in the mutant or small molecule treated Zebrafish, and conversely to determine whether global inhibition could rescue the Lefty knockout phenotype in gastruloids.

While Lefty feedback is not required for patterning in Zebrafish, it is required for scaling the size of the embryo[41]. However, mammalian gastruloids do not scale[23,42] and early mammalian embryos respond to loss of cells through regrowth, not by changing the scale of the patterning. It is possible that these functional differences in patterning correspond to differences in the Nodal-Lefty system, and that Lefty is therefore required for patterning in mammals not only for robustness or scaling. Whether this is the case remains an open question. Taken together, our study shows that Nodal propagates through a transcriptional relay whose timing is controlled by Lefty, and therefore defines a previously unappreciated mechanism of patterning by an activator-inhibitor pair that operates in specifying cell fates in human 2D gastruloids.

## Methods

### Cell line

*Human embryonic stem cell (hESC) line.* The ESI-017 hESC line used in this study was purchased from ESIBIO. Karyotype and genomic integrity check were done by the manufacturer. Pluripotency (OCT4+, SOX2+, NANOG+) was regularly monitored prior to and during the study. Nodal knockout cells were previously generated and characterized in this lab[28]. GFP- β-catenin cells were previously generated and characterized in this lab[43]. All cell lines were tested regularly for mycoplasma and found negative.

All experiments were performed with human embryonic stem cells lines that were previously established and are in the NIH registry. Experiments conform to the ISSCR guidelines found at https://www.isscr.org/policy/guidelines-for-stem-cell-research-and-clinical-translation.

**Cell culture**. hESCs were maintained in serum-free, mTeSR1 (STEMCELL Technologies) medium, in Matrigel (Corning; 1:200 in DMEMF12) coated culture dishes, at 37 °C, 5% carbon dioxide ($CO_2$). mTeSR Plus (STEMCELL Technologies) was used for maintaining cNodal and cNodal Lefty compound knock-out cells shown in Fig. 6b. Essential 6 medium (E6) (Gibco) was used for TGF-ß ligand-free conditions. All experiments were performed with cells not exceeding passage number 60. Dispase (Fisher Scientific) for gentle dissociation was used for routine maintenance. Accutase (Corning) was used to prepare single cell suspensions. ROCK-inhibitor Y27672 (10 μM; Stem Cell Technologies) was used to support single cell viability.

Antibiotics were used for selection. In particular, 100 μg/ml G418 or 5 μg/ml Blasticidin was used to select cells with a modified Nodal locus; 1 μg/ml Puromycin was used to enrich for cells transiently transfected with px459 plasmids; 5 μg/ml Puromycin was used for selecting the mCherry-CAAX cell membrane marker. 5 μg/ml Blasticidin was used for the CFP-H2B nuclear marker. Inducible expression experiments were performed in mTeSR medium supplemented with the indicated factors for the indicated times.

**Plasmids**. Plasmids were constructed for this study as follows.

1) ePiggyBac transposable element[44] vectors for stable integration of genes of interest. An ePiggyBac master vector based on the pBSSK backbone, harboring transposon-specific inverted terminal repeat sequences (ITRs) was modified to deliver: nuclear marker CFP-H2B (Plasmid AW-P28), data shown in Fig. 1c; doxycycline inducible cell membrane marker mCherry-CAAX (Plasmid AW-P224), data shown in Figs. 2, 3. To construct a CitrineTrap (Morphotrap) expression vector (Plasmid AW-P227), the full length of the GFP-Trap coding sequence was subcloned from plasmid 1134-morphotrap (a gift from Dr. Markus Affolter, University of Basel) and inserted into ePiggyBac master vector under the control of the doxycycline inducible TRE promoter.

2) CRISPR/Cas9 vectors for genome-editing. To generate insertions (cNodal, cEcad) or nonsense mutations (Lefty 1/2), a Cas9 and sgRNA co-expression vector, either px459 (Addgene) or px330 (Addgene), was modified to target the locus of interest: Nodal mature domain targeting plasmid AW-P188, derived from px330, sgRNA spacer sequence 5′-TGTCTGGCAAGTGATGTCGA-3′; Lefty 1 exon1 targeting plasmid AW-P219, derived from px459, sgRNA spacer sequence 5′-GCAGCACCATGCAGCCCCTG-3′; Lefty 2 exon1 targeting plasmid AW-P220, derived from px459, sgRNA spacer sequence 5′- GCAGCACCATGTGGCCCCTG-3′. CDH1 (E-cadherin) Exon16 targeting plasmid AW-p191, derived from px330, sgRNA spacer sequence 5′-TGACATGTACGGAGGCGGCG-3′. All sgRNA sequences used in this study were designed using the Benchling built-in function Design CRISPR Guide. Off-target score was evaluated using the guidelines at crispr.mit.edu (TOOLS FOR GUIDE DESIGN > Benchling) from[45]. On-target score was evaluated according to the optimized score from[46], only available for SpCas9. In all cases, sgRNA sequence was inserted into px330 or px459 using BbsI sites.

3) Plasmids for homology directed DNA repair. Two donor vectors were created for biallelic Nodal locus insertion. The configuration of the two vectors is shown in Supplementary Fig. 1A. In the FloxP-neomycin resistant vector (AW-P216), 800 bp of Nodal pro-domain or Nodal mature domain sequence was used as right or left homology arm, respectively, flanking the Nodal-neomycin expression cassette and mCitrine. In the FloxP-blasticidin-2A-RFP vector (AW-P218), 800 bp of Nodal pro-domain was used as right-arm, and 450 bp of Nodal mature domain was used as left-arm. Cre expression plasmid (Cre Shine, Addgene #37404) was used for excision of FloxP fragments after antibiotic-resistance selection. A plasmid (AW-P133) previously created in this lab, harboring mCitrine and a resistance gene (Blasticidin S deaminase (BSD) co-expression cassette), was used as PCR template to generate linear DNA donor flanked by 45 bp arms homologous to CDH1 Exon16 sequence near the stop codon to generate E-cadherin-mCitrine fusion.

PCR for DNA fragment sub-cloning for plasmid DNA construction was done using OneTaq Hot Start (NEB), Phusion Hot Start (NEB) or Q5 (Fisher Scientific) DNA polymerase, following the manufacturer's instructions with optimized annealing temperature. Optimized conditions were used for Nodal modified cells genotyping PCR[47].

**DNA nucleofection and stable cell line establishment**. DNA nucleofection was performed with the 4D-Nucleofector (Lonza) system according to the manufacturer's protocol using the P3 Primary Cell 4D-Nucleofector Kit (Lonza). For ePiggyBac transposase mediated insertion, antibiotic selection started on day 2 (two days after nucleofection), and lasted at least for 7 days. For CRISPR/Cas9 mediated knockout, px459 plasmid transfectants was selected on day 1, for one day. For CRISPR/Cas9 mediated insertion, donor DNA was linearized and concentrated before nucleofection. Selection started on day 3, and lasted for 7–10 days, until no cell death and healthy colonies were established.

For ePiggyBac transposase mediated insertion, cells were pooled after selection. Corresponding antibiotics were not supplemented in medium for routine culture, until 2–3 days prior to an experiment. For CRISPR/Cas9 mediated knockout, single clones were handpicked and amplified. Sub-cloning was conducted as needed. Sanger sequencing was performed to screen retrieved colonies. Promising knockout candidates were further confirmed by sequencing individual alleles with TOPO-cloning (Invitrogen). Only confirmed knockout mono-clones were used for further experiments. For CRISPR/Cas9-mediated insertion of mCitrine to Nodal locus, the insertion procedure was done for twice to create the homozygous knock-in line. The modified locus is not recognized by the sgRNA so the first knock-in allele was not edited in producing the second. Single-cell Fluorescence-activated cell sorting (FACS) was conducted to isolate desired mono-clones, with correct genomic modification. After establishment, stable lines were checked for pluripotency markers, i.e., OCT4, SOX2 and NANOG expression, and found indistinguishable from WT ESI-017.

**Single-cell FACS**. We found that transiently expressed Cre-mediated excision of FloxP fragments inserted in Nodal locus in hESCs was extremely inefficient. Therefore, desired clones with FloxP excised could only be isolated by single-cell FACS after excision. A SH800S Cell Sorter (Sony) equipped with Cell Sorter software (version 2.2.4.5150) was used to sort single cells into individual wells of 96-well plates. Potential heterozygotes were treated with 10 ng/ml Activin for 6 h in order to increase cNodal expression, so that mCitrine could be used as marker for FACS. Potential homozygotes were also pre-treated with Activin and sorted as mCitrine-positive/RFP-negative. Sorted single cells were kept in mTeSR medium

supplemented with 1x CloneR (STEMCELL Technologies), to support viability and genomic integrity. Mono-clones were established and characterized by genotyping PCR and Sanger sequencing (shown in Supplementary Fig. 1).

**Micropatterned hESCs-based gastruloids**. We used CYTOO plates 96 DC-S-A glass-bottom 96-well microplates with circular micropatterns (700 μm diameter) for creation of gastruloids. On the day of seeding cells, micropatterns were coated with 200 μl of 5 μg/ml laminin-521 solution (LN521, Biolamina) in DPBS (with calcium and magnesium, Lonza) for 2 h at 37 °C. Unattached laminin was washed out with 200 μl of DPBS, repeated 4 times.

hESCs were passaged using dispase 2–3 days prior to micropatterned cell seeding and maintained in mTeSR medium. By the day of seeding, cells reached 50–70% confluency, with the majority of the colonies between 500 and 1000 μm in diameter. Single cell suspension was prepared using Accutase, and cells were counted using a hemocytometer. Unless specifically indicated, about 150,000 cells/ 150 μl mTeSR were placed into each well of the laminin coated micropattern plate and incubated at 37 °C for 45 min to allow the cells to attach to micropattern surface. Then, unattached cells were washed away with PBS twice, before adding induction medium. The induction medium is mTeSR supplemented with 50 ng/ml BMP4, unless otherwise specified. For detailed protocol, refer to[48].

**Juxtaposition analysis**. For cNodal and Lefty protein diffusivity analysis, homozygous cNodal hESCs or Lefty 1 and Lefty 2 (Lefty1/2) compound knockout hESCs (for background assessment) were used as senders. Lefty1/2 compound knockout hESCs were used as receivers. For temporal analysis experiments, sender cells were dissociated with Accutase and 100,000 cells were resuspended in 50 μl mTeSR supplemented with ROCK-inhibitor Y27672 (10 μM) and seeded in one well of 2 well silicone insert with 0.22 cm² growth area (ibidi), in a Matrigel coated μ-Slide 8 Well (ibidi), and then incubated at 37 °C for 1–2 h to allow the cells to attach to the culture surface. Unattached cells were washed out with PBS twice. Sender cells were then induced with 10–50 ng/ml Activin in mTeSR for 20 h. The following day, Lefty1/2 compound knockout cells were dissociated with Accutase and 200,000 cells were resuspended in 100 μl mTeSR supplemented with ROCK-inhibitor Y27672 (10 μM), and seeded in the same well, juxtaposed with sender cells. After 30 min, unattached receiver cells were washed out with PBS. Then, 150 μl mTeSR supplemented with or without 50 ng/ml Activin was added into the well. Cells were fixed at indicated time points for further assessments of cNodal and Lefty protein.

For positive autoregulation-dependent relay analysis, WT hESCs or homozygous cNodal hESCs were used as positive sender cells, Nodal knockout hESCs were used as negative sender cells. WT hESCs or Nodal knockout hESCs were used as receiver cells. For assays under stationary conditions, sender cells were seeded in a silicone insert and kept in mTeSR supplemented with 10 ng/ml Activin for 16–20 h. Prior to seeding of receiver cells, sender cells were washed twice with PBS, and then once with mTeSR or E6. Specific receiver cells were seeded juxtaposed with sender cells. Cells were kept in media alone (mTeSR or E6), and then were fixed at the indicated times for further analysis of Smad2/3, Nodal transcripts or Lefty transcripts. Of note, to minimize basal levels of Nodal expression caused by TGF-ß in mTeSR, the Nodal FISH experiment shown in Fig. 2F was done with sender and receiver cells kept in E6 medium. In addition, 10 μM SB431542 was added into the medium at 1 h prior to dissociation of receiver cells for seeding. In that case, the receiver cells were washed two times with E6 after accutase digestion to remove SB431542.

Heterogeneous or decreased Nodal expression or signaling levels were occasionally observed in the sender cell colony center after long incubation times (after 24–48 h h), which might be caused by high cell density[32]. To avoid these effects complicating the analysis, freshly seeded sender cells were induced with Activin for 16–20 h or relatively low-density cells at the colony edge of sender cell colonies were imaged as signal sources.

**Single molecule fluorescence in situ hybridization (smFISH)**. Cells were fixed in 4% paraformaldehyde for 15 min. smFISH was performed following manufacturer's instructions (ACD Bio, RNAscope fluorescent multiplex assay), with indicated probes (Supplementary Table 1). Cells were placed in PBS before imaging.

**Immunofluorescence staining**. Cells were fixed in 4% paraformaldehyde for 15 min at room temperature. Fixed samples were then washed with PBS twice before permeabilization and blocking with 3% Donkey serum in PBST (1 x PBS with 0.1% Triton X-100) for 30–60 min at room temperature. For pSmad2/3 staining, cells were incubated in PBS with 1% SDS for 30 min at 37 °C following blocking. Primary antibodies were diluted in blocking buffer as indicated (Supplementary Table 1). Cells were incubated with diluted primary antibody solution at room temperature for 1 h or at 4 °C overnight, washed three times with PBST for 30 min each time, and then incubated with 1:500 diluted secondary antibodies (Supplementary Table 1) solution supplemented with DAPI at room temperature for 1 h, and then washed with PBST 3 times. Cells were placed in PBS before imaging.

For dual smFISH/immunofluorescent labeling, the smFISH procedure was performed first and then cells were washed with PBS and incubated with primary

antibodies. The remainder of the immunofluorescence procedure specified above was then followed.

### Imaging

*Live cell imaging.* Cells seeded in μ-Slide 18 well or 8 well (ibidi) chamber slides or CYTOO micropattern 96-well plates were imaged on Olympus/Andor spinning disk confocal microscope equipped with environmental chamber, with a 20x, NA 0.75 objective or a 40x, NA 1.25 silicon oil objective. Images were acquired and exported from Andor iQ3 software. During imaging, temperature (37 °C), humidity (~50%), and $CO_2$ (5%) were controlled. 4 positions of each condition were selected for imaging.

*Fixed cell imaging.* Fixed micropatterned cells were imaged on Olympus/Andor spinning disk confocal microscope, using 20x, NA 0.75 objective, with Andor iQ3 software. 6 to 10 colonies were imaged for each condition and 4 positions were imaged for each colony so that the entire colony could be covered and reconstituted via stitching; Olympus IX83 inverted epifluorescence microscope with CellSens Dimension (1.18) software, and Olympus FV1200 laser scanning confocal microscope (LSM) with FV10-ASW 4.2 software and 20x, NA 0.75 objective were also used for fixed micropattern imaging. Fixed cells in chamber slides were imaged on a spinning disk confocal microscope with a 20x, 0.75 NA or a 40x, NA 1.25 silicon oil objective or on an epifluorescence microscope with a 20x, 0.75 NA objective. 4 positions were imaged for each condition.

**Image analysis**. Imaging experiments were repeated at least twice with consistent results, 'n' in the figure captions denotes the number of micropattern colonies in the same experiment. Completely filled micropatterned colonies were selected for imaging based on visualization of the brightfield or DAPI channels without examining the other channels to avoid any biases in colony selection. Images obtained from various experimental conditions were processed using Fiji[49] to set lookup tables for visualization. Filters (Remove outliers) were applied to some raw images to remove non-specific bright puncta.

Analysis was performed using custom MATLAB code, available at https://github.com/warmflashlab/Liu2021. For analysis of 2D gastruloids, micropatterned colonies were automatically identified and the marker of interest quantified at each point in space. Averages were than taken over the angular coordinate for each colony to produce the average intensity as a function of the radial coordinate for that colony in 10 μm bins. These were then averaged over all colonies for that condition and the means and standard errors of measurement displayed in the figure.

For live cell imaging of cNodal cells, cells were segmented using custom MATLAB code and average cNodal fluorescence over both nucleus and cytoplasm was computed for each cell and then these values were averaged over all cells for each field of view. The plots show mean and standard error of measurement across at least five fields of view.

For juxtaposition experiments, DAPI channel images were first segmented via Ilastik[50] to detect all nuclei. The channel representing one of the two cell types (usually the sender cells) was also segmented. Based on this, binary image masks were created to contain either only producing or only receiving cell areas. Images were background-subtracted, and protein expression was quantified in receiving cells as a function of distance from producing cells in several-micron increment bin away from the border between cell types. Distance from the border was determined using the MATLAB function *bwdist* applied to producing cells' mask. smFISH images were quantified in similar way. To quantify smFISH in standard culture (Supplementary Fig. 10a) mean pixel intensity was quantified from masks generated via cell nuclei segmentation and dilation, in order to account for cytoplasmic mRNA molecules. Error bars represent SEM over multiple images at each time point. Kymographs of micropatterned images (in Fig. 1c) were obtained as averages over three circular colonies containing four pie-segments each, i.e., averaged over 12 pie segments for each treatment condition.

**Statistical analysis**. Multiple images of live-cell fields or micropatterns or juxtaposition cultures or FISH experiments contributed to each experimental condition. In all cases, error bars represent standard error of the mean.

**Reporting summary**. Further information on research design is available in the Nature Research Reporting Summary linked to this article.

## Data availability

Source Data are provided with this paper. Data of individual micropattern colonies describing variability of the intensity are provided as Supplementary Figures. Immunofluorescence and live-cell imaging raw data underlying quantification are provided (https://doi.org/10.6084/m9.figshare.17420450.v1). Plasmid DNA used in this study will be available from the corresponding author upon reasonable request. Source data are provided with this paper.

## Code availability

Analysis was performed using MATLAB (2020b) as described in the "Methods" section. Custom code for image analysis is available at https://github.com/warmflashlab/Liu2021.

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

## Acknowledgements

We thank Lauren Gambill and Ramya Ganiga Prabhakar for FACS technical support. We also thank Eric Siggia, Elena Camacho Aguilar, Xiangyu Kong and Miguel Ángel Ortiz Salazar for comments. We thank Markus Affolter for sharing GFP-Trap plasmid. This work was funded by Rice University and grants from the Welch Foundation (C-2021), NSF (MCB-1553228), NIH (R01GM126122), and Simons Foundation (511079).

## Author contributions

A.W., L.L., and I.H. conceived the project. L.L. and A.W. designed experiments. L.L., S.C., A.W., and I.H. contributed to methodology. L.L., L.R., J.Y.J., and M.C.G. performed experiments. L.L., A.N., and S.C. analyzed the data, with supervision of A.W. Illustrations were prepared by L.L and A.N. A.W. supervised the project and secured funding. The manuscript was drafted by L.L and A.W. with input from A.N., S.C., and I.H.

## Competing interests

The authors declare no competing interests.
