## [Peer Review File · Nature Communications]

Nodal is a short-range morphogen with activity that spreads through a relay mechanism in human gastruloidsREVIEWER COMMENTS

Reviewer #1 (Remarks to the Author):

This extremely exciting study of Liu et al. is a major advance for the field of early patterning by Nodal morphogen signaling for multiple reasons. It visualizes for the first time endogenous NODAL protein and its signaling range, and in a model of human embryonic development, and its modulation by endogenous Lefty proteins. The conclusions are important and very interesting, and I find them to be well substantiated, aside from a few questions mentioned below about statistics. The experimental designs are extremely elegant (including the nanobody trap for secreted mCitrine-Nodal fusion protein in signal receiving cells, which is another game changer in the field). I enthusiastically recommend publication of this manuscript. I only have few points that I think should be addressed.

1) One important question is to what extent the mCitrine tag might alter NODAL activity. The authors convincingly show that it does not prevent mesoderm induction in 2D gastruloids, and even in the absence of endogenous untagged NODAL. They quantified the mesoderm marker BRA by IF staining (Fig. 1d). However, in the top row images, they should include the single channel fluorescence images and indicate at which positions they quantified the fluorescence intensities.

2) Are the color codes inverted in the first panel (top row) of Fig1d? Or why else would Sox2 stain at the periphery rather than at the center in the first panel (Nodal +/- control)? Or of the color coding is the same across all three panels, why does the distribution of Sox2 protein change when Nodal is tagged with mCitrine?

3) Fig. 5b,c shows an expansion of BRA+ mesoderm at the expense of SOX2+ ectoderm in L1/2 KO, as expected if Lefty1,2 increases the threshold of NODAL required to induce mesoderm. However, how did the authors validate that this change is statistically significant? And when they say n=6, are these repeat experiments, or 6 colonies from the same experiment? And if they selected only 6 gastruloids, how can the reader be sure that these were not selected to fit the prediction?

4) Interestingly, no corresponding expansion was observed for SOX17+ endoderm. According to one existing model, endoderm formation requires prolonged NODAL signaling. What if they would stain SOX17 in their WT and L1/2 KO gastrulation at a later time point at 54 or 72 hrs? And did the authors consider to introduce the CitrineTrap nanobody into L1,2 KO cells? It would probably allow to directly measure the influence of LEFTY proteins on the rate of Nodal protein diffusion in their cell juxtaposition assay (if needed with the help of FRAP assays) – a parameter which in turn would be key for mathematical modeling of gradient formation by endogenous NODAL protein.

5) An important and very interesting difference is reported in the spread of Lefty proteins in cell juxtaposition assays versus 2D gastruloids. Could they comment in the Discussion possible mechanisms that could potentially account for this difference?

Minor points:

The legend for Fig. 2b seems to lack a verb in the last sentence.

Fig.S3a legend: Wide-type should be wild-type

Images in Fig. S5 lack size bars.

How does the quantification in Fig. 4c account for variability among the (six?) replicates analyzed?

Reviewer #2 (Remarks to the Author):

This is a characteristically elegant, thoughtful, technically sound and important piece of work from the Warmflash lab. The experiments address the issue of the mechanism of Nodal signalling and, in particular, its relationship of its spatial organization using micropatterned hESCs. The approach to the question relies on a series of elegant reagents involving Nodal, a reporter fusion protein with full Nodal activity and a Nodal nanobody based trap.

There is a well established view that Nodal and Lefty are part of a reaction diffusion system that mediates its spatial signalling dynamics. The authors have discussed earlier this in a different context and concluded that this notion, mostly but not uniquely, derived from work with zebrafish does not apply to mammalian signalling. Work in zebrafish, duly cited in the text, had already questioned Nodal long range signalling and reaction diffusion but a proper mechanism had been missing. Here, the authors conclude that, in this system, Nodal diffusion is short-range and its transmission likely to rely on a cell-cell contact mechanism where Nodal induces its own expression in neighbouring cells and signal transmission relies completely on Nodal activation in receiver cells. To prove the hypothesis they present a set of 4 juxtaposition experiments where receiver cells are either WT or Nodal KO cells. They could observe pSmad2/3 in the receiver compartment only when receiver cells were WT, hence transmission of the signal is obtained only when all the cells can produce their own Nodal. When the receivers are Nodal KO, Smad2/3 is not phosphorylated and stays in the cytoplasm. The data is compelling and their conclusion supported by elegant experiments with a Nanobody trap for Nodal. Having established this, the authors address the role of Lefty on Nodal signalling, concluding that it regulates its spread dynamics and not necessarily its spatial organization.

The work is technically sound and the conclusions well supported by the experiments and the data. The work is important because it challenges a widespread assumption about Nodal signalling in embryos. In this regard, it would be good to see here a further development of the argument against a Turing mechanism that the authors have presented elsewhere. Perhaps they can have an updated discussion in the supplementary materials.

Two details

In Fig.1d top right it is possible that Brachyury and Sox2 have been mislabelled by accident, as their localisation in the gastruloid contradicts what stated elsewhere in the paper and in previous publications.

Also, the use of the term gastruloid for the micropatterns is confusing and, as they do in some places, it might be good to refer to them as either 2D-gastruloids, as they have in some recent publications, or M-gastruloids for micropatterned gastruloids. In any event, this is a minor remark that the authors can ignore, though it can cause confusion in a field with too many acronyms and names.

REVIEWER COMMENTS

Reviewer #1 (Remarks to the Author):

This extremely exciting study of Liu et al. is a major advance for the field of early patterning by Nodal morphogen signaling for multiple reasons. It visualizes for the first time endogenous NODAL protein and its signaling range, and in a model of human embryonic development, and its modulation by endogenous Lefty proteins. The conclusions are important and very interesting, and I find them to be well substantiated, aside from a few questions mentioned below about statistics. The experimental designs are extremely elegant (including the nanobody trap for secreted mCitrine-Nodal fusion protein in signal receiving cells, which is another game changer in the field). I enthusiastically recommend publication of this manuscript. I only have few points that I think should be addressed.

We thank reviewer #1 for their positive comments and constructive criticism. We address the individual points below.

1) One important question is to what extent the mCitrine tag might alter NODAL activity. The authors convincingly show that it does not prevent mesoderm induction in 2D gastruloids, and even in the absence of endogenous untagged NODAL. They quantified the mesoderm marker BRA by IF staining (Fig. 1d). However, in the top row images, they should include the single channel fluorescence images and indicate at which positions they quantified the fluorescence intensities.

Response:

Single channel images are now provided.

Fluorescence intensities were not measured at only one particular point in the colony but instead the entire colony was computationally identified and then the intensity computed for each point. This two-dimensional intensity field was then averaged over the angular coordinate for each colony to give an average as a function of the radial coordinate in bins of 10 μm . These averages for each colony were then averaged over 10 colonies to give the plots shown. We also show the standard error of measurement across the 10 colonies. We have now expanded the description of the image analysis in the methods section to clarify these points.

2) Are the color codes inverted in the first panel (top row) of Fig1d? Or why else would Sox2 stain at the periphery rather than at the center in the first panel (Nodal +/+ control)? Or of the color coding is the same across all three panels, why does the distribution of Sox2 protein change when Nodal is tagged with mCitrine?

Response:

We apologize for this error - the color codes were accidentally switched. The figure has now been corrected.

3) Fig. 5b,c shows an expansion of BRA+ mesoderm at the expense of SOX2+ ectoderm in L1/2 KO, as expected if Lefty1,2 increases the threshold of NODAL required to induce mesoderm. However, how did the authors validate that this change is statistically significant? And when they say n=6, are these repeat experiments, or 6 colonies from the same experiment? And if they selected only 6 gastruloids, how can the reader be sure that these were not selected to fit the prediction?

Response:

The data shown in Fig 5b, c, represent 6 colonies from the same experiment, which were selected based on the DAPI channel showing typical colony morphology and cell density. As colonies were selected based on the morphology from the DAPI channel before the BRA image was acquired, they were not biased to only include those with a specific profile. Quantification was performed by automated MATLAB code in an unsupervised manner. We now present error bars (s.e.m) across the six colonies in Figure 5 which demonstrate that the differences are highly significant. We revisited data from four experiments to verify the phenotype is reproducible between experiments and we also created a Nodal-Citrine Lefty1/2 knockout cell line to further verify this phenotype, the result is shown in fig 5c and in Figure R1. In Figure R1, we also show scans of larger regions of the culture dishes to show that the phenotype is reproducible across all colonies. We have also expanded the details of the analysis in methods to clarify these points.

Figure R1. Depletion of Lefty1/2 leads to expansion of Nodal expression and mesodermal fate (A), Means and standard error (left) and individual colony quantification (right) of the results shown in Fig. 5b. (B) Comparison of Nodal^{Cit/Cit} cells and Nodal^{Cit/Cit} Lefty double knockout cells. (Top) live-cell imaging show the cNodal expression pattern at 24h post-BMP4, 6 adjacent colonies were scanned. (Bottom) Immunostaining of cells at 37h-post BMP4, the entire micropattern areas with 25 colonies in each well were scanned. cNodal (anti-GFP), SOX2 and BRA were checked, scale bar, 600 μ m. (C) 6 colonies in (B) with similar morphology are quantified.

4) Interestingly, no corresponding expansion was observed for SOX17+ endoderm. According to one existing model, endoderm formation requires prolonged NODAL signaling. What if they would stain SOX17 in their WT and L1/2 KO gastrulation at a later time point at 54 or 72 hrs? And did the authors consider to introduce the CitrineTrap nanobody into L1,2 KO cells? It would probably allow to directly measure the influence of LEFTY proteins on the rate of Nodal protein diffusion in their cell juxtaposition assay (if needed with the help of FRAP assays) – a parameter which in turn would be key for mathematical modeling of gradient formation by endogenous NODAL protein.

Response:

A recent preprint has shown most SOX17 positive cells in micropatterns at the time we consider are primordial germ cells (PGCs) while more endoderm emerges later. Moreover, it showed that while Nodal signaling is required for the emergence of PGCs, there is no dose-dependent enhancement of PGCs when Nodal signaling is further activated. Thus, it is possible that a baseline level of PGC differentiation is masking the differences in endoderm levels. That same study also shows that the peak of the SOX17+ endodermal emergence is at significantly later time points than we examine in our study (72h vs 42 hrs). We note that while the spatial range of SOX17+ cells remains the same, there is an increase in Sox17 fluorescence intensity which may result from a mild increase in endoderm differentiation at the time points we examine. The effect of Lefty KO on endoderm differentiation is an interesting question for future study, however, we believe that because of the confounding factors involved and the differing timing, this is beyond the scope of our present study. We now cite this preprint and explain that SOX17 marks both PGCs and endoderm (*Jo, K. et al. Efficient differentiation of human primordial germ cells through geometric control reveals a key role for NODAL signaling. bioRxiv 2021.08.04.455129 (2021) doi:10.1101/2021.08.04.455129.*)

As suggested, we attempted to test whether the citrine trap could mimic the effects of Lefty by introducing it into cNodal;L1^{-/-};L2^{-/-} hESCs, however, this experiment showed that binding to the morphotrap appears to stabilize the cNodal protein increasing its total level, while it remained functional, increasing the levels of pSmad2/3. Thus, there is a crucial difference between Lefty and morphotrap in that Lefty proteins inactivate Nodal while morphotrap does not. These data are reproduced below, Figure R2, but not included in the manuscript. We also note that these experiments are consistent with our picture of exclusively short-range Nodal transport in that introducing morphotrap did not appear to limit the spread of Nodal but instead it covered the same range but was actually upregulated due to stabilization. This supports the idea that in any event Nodal only moves at short range and so expression of a morphotrap does not affect its spatial range.

Figure R2. Morphotrap stabilizes but does not inactivate cNodal. (A) Representative images of BMP4 treated homozygous cNodal cells (H3), cNodal cells with Lefty double knockout, and cNodalTrap. Experiments repeated twice with consistent results. (B) Quantification of 6 colonies from each group in the above experiment. BRA levels are normalized to DAPI.

5) An important and very interesting difference is reported in the spread of Lefty proteins in cell

juxtaposition assays versus 2D gastruloids. Could they comment in the Discussion possible mechanisms that could potentially account for this difference?

Responses:

We believe there are three possible, not mutually exclusive, explanations for this difference. (1) It could be a difference of time scale: while stable expression in the juxtaposition assays allows for a gradient of Lefty to form, the rapidly evolving patterns in gastruloids do not. (2) It is possible that greater accumulation of extracellular matrix or other factors limit Lefty spread in the micropatterned assay, and (3) It is possible that the relative expression of Lefty1 and Lefty2 differs between the two assays. Our immunostaining results with Lefty 1 or Lefty 2 knock-out cells suggest that both Lefty 1 and 2 are expressed, however due to the similarity of Lefty 1 and Lefty 2 in mRNA and amino acid sequences, neither the FISH probes nor antibody can distinguish the Lefty 1 and 2 expression pattern in space and time. Interestingly, our unpublished data indicate that mammalian Lefty1 and 2 may have differential diffusivity, specifically, Lefty 2 shows limited diffusivity in juxtaposition setting while Lefty1 diffuses at long range as seen for total Lefty. One possible scenario is that in 2D gastruloids, Lefty 2 might be the dominant one, and if that is true, it would be consistent with the posteriorized nature of our 2D gastruloids. Mouse embryo studies show Lefty2 is mainly expressed at posterior end of early gastrulation stage embryo. Although inclusion of this unpublished data is beyond the scope of the manuscript, we now expand the discussion of these points to include these three possibilities.

Minor points:

The legend for Fig. 2b seems to lack a verb in the last sentence.

Fig.S3a legend: Wide-type should be wild-type

Images in Fig. S5 lack size bars.

We thank the reviewer for noticing these issues which have now been corrected.

How does the quantification in Fig. 4c account for variability among the (six?) replicates analyzed?

Response: the line plots show the mean value extracted from 6 colonies in the same experiment. We now also include error bars indicating the s.e.m., and individual colonies in Fig S8. Additionally, we have clarified this point in the figure caption.

Reviewer #2 (Remarks to the Author):

This is a characteristically elegant, thoughtful, technically sound and important piece of work from the Warmflash lab. The experiments address the issue of the mechanism of Nodal signalling and, in particular, its relationship of its spatial organization using micropatterned hESCs. The approach to the question relies on a series of elegant reagents involving Nodal, a reporter fusion protein with full Nodal activity and a Nodal nanobody based trap.

There is a well established view that Nodal and Lefty are part of a reaction diffusion system that mediates its spatial signalling dynamics. The authors have discussed earlier this in a different context and concluded that this notion, mostly but not uniquely, derived from work with zebrafish does not apply to mammalian signalling. Work in zebrafish, duly cited in the text, had already questioned Nodal long range signalling and reaction diffusion but a proper mechanism had been missing. Here, the authors conclude that, in this system, Nodal diffusion is short-range and its transmission likely to rely on a cell-cell contact mechanism where Nodal induces its own expression in neighbouring cells and signal transmission relies

completely on Nodal activation in receiver cells. To prove the hypothesis they present a set of 4 juxtaposition experiments where receiver cells are either WT or Nodal KO cells. They could observe pSmad2/3 in the receiver compartment only when receiver cells were WT, hence transmission of the signal is obtained only when all the cells can produce their own Nodal. When the receivers are Nodal KO, Smad2/3 is not phosphorylated and stays in the cytoplasm. The data is compelling and their conclusion supported by elegant experiments with a Nanobody trap for Nodal. Having established this, the authors address the role of Lefty on Nodal signalling, concluding that it regulates its spread dynamics and not necessarily its spatial organization.

We thank reviewer #2 for their positive comments on our work and respond to individual issues below.

The work is technically sound and the conclusions well supported by the experiments and the data. The work is important because it challenges a widespread assumption about Nodal signalling in embryos. In this regard, it would be good to see here a further development of the argument against a Turing mechanism that the authors have presented elsewhere. Perhaps they can have an updated discussion in the supplementary materials.

Response: We have now expanded the discussion of this point in the Discussion section.

Two details

In Fig.1d top right it is possible that Brachyury and Sox2 have been mislabelled by accident, as their localisation in the gastruloid contradicts what stated elsewhere in the paper and in previous publications.

Response: We thank the reviewer for point this out which was indeed an error. It has now been corrected.

Also, the use of the term gastruloid for the micropatterns is confusing and, as they do in some places, it might be good to refer to them as either 2D-gastruloids, as they have in some recent publications, or M-gastruloids for micropatterned gastruloids. In any event, this is a minor remark that the authors can ignore, though it can cause confusion in a field with too many acronyms and names.

Response: We agree with the reviewer and now refer to these as 2D gastruloids throughout.

REVIEWERS' COMMENTS

Reviewer #1 (Remarks to the Author):

The authors have addressed all of my points.

Reviewer #2 (Remarks to the Author):

The authors have responded to the reviews and increase the already high quality of the work.